# A Sec-Dependent Effector from “*Candidatus* Phytoplasma ziziphi” Suppresses Plant Immunity and Contributes to Pathogenicity

**DOI:** 10.3390/biology14050528

**Published:** 2025-05-10

**Authors:** Cui-Ping Wan, Fu-Xin He, Wei Zhang, Qian Xu, Qi-Liang Zhu, Chuan-Sheng Song

**Affiliations:** 1College of Agricultural and Biological Engineering, Heze University, Heze 274015, China; 17615621665@163.com (W.Z.); 17615533505@163.com (Q.X.); zql212linxue@163.com (Q.-L.Z.); 2Shanxi Key Laboratory of Integrated Pest Management in Agriculture, College of Plant Protection, Shanxi Agricultural University, Jinzhong 030600, China; hefuxin@sxau.edu.cn

**Keywords:** *Candidatus* Phytoplasma ziziphi, phytoplasma, *Ziziphus jujuba*, Sec-dependent effectors, reactive oxygen species, callose deposition

## Abstract

Jujube witches’ broom (JWB) disease, caused by a phloem-limited phytoplasma, severely damages jujube tree production. This study investigates how a specific pathogen effector, JWB790, compromises plant defenses to promote disease. Experiments showed that JWB790 is abundant in JWB-infected jujube and suppresses plant-innate immune responses, including programmed cell death, H_2_O_2_ accumulation, and callose deposition. Transgenic *Arabidopsis thaliana* plants overexpressing *JWB790* exhibited enhanced pathogen susceptibility. Genetic analysis revealed that JWB790 alters the expression of biotic stress-related genes. These findings demonstrate that JWB790 acts as a key virulence factor, suppressing plant immunity and facilitating pathogen dissemination. Understanding how JWB790 works provides critical insights into combating JWB disease, which could help protect jujube crops and support agricultural sustainability.

## 1. Introduction

Jujube (*Ziziphus jujuba* Mill.), an economically significant fruit tree in China, is highly valued for its drought tolerance, ecological adaptability, and ornamental benefits [1]. However, jujube witches’ broom (JWB) disease, caused by the phloem-restricted pathogen *Candidatus* Phytoplasma ziziphi (*Ca*. P. ziziphi), poses a devastating threat to jujube cultivation [2]. Characteristic symptoms of the disease include the formation of witches’ broom (stem proliferations) and phyllody (the retrograde development of flowers into vegetative tissues), ultimately leading to the death of the tree within several years [3]. *Ca*. P. ziziphi is a member of the elm yellows group (16SrV-B), possessing a highly reduced genome of approximately 760 kb [4,5]. Notably, JWB phytoplasmas lack a set of genes critical for essential metabolic pathways, rendering them obligate parasites that rely entirely on their jujube host for nutrition and making in vitro cultivation exceptionally challenging. Therefore, investigating the battle strategies between *Ca*. P. ziziphi and jujube plants is critical for developing control measures.

In the co-evolutionary arms race between plants and pathogens, plants have evolved sophisticated multilayered immune systems to detect and counteract pathogen invasion [6,7]. The primary defense layer involves plant pattern recognition receptors (PRRs), which perceive conserved pathogen-associated molecular patterns (PAMPs), thereby activating PAMP-triggered immunity (PTI) to restrict microbial colonization [8]. To establish successful infections, adapted pathogens deploy an arsenal of secreted effectors to suppress or subvert PTI signaling [9]. The pathogenic mechanisms by which oomycete, flagellate fungi, and bacterial effectors suppress immunity have been extensively studied [10,11], while those of phytoplasma effectors remain poorly understood. Phytoplasma genomes harbor a functional Sec-dependent protein translocation pathway (i.e., SecA, SecE, and SecY), which transports proteins to the exterior of the sieve elements where the phytoplasmas reside [5]. Sec-dependent effectors (SDEs) are of great importance for the pathogens like phytoplasmas to infect plants and insects [12,13]. However, our knowledge regarding the function of how SDEs inhibit plant immunity is still rather limited.

The transient expression of effectors in *Nicotiana benthamiana* is widely used for the large-scale screening of virulence factors, especially those of obligate parasites [14]. Hypersensitive response (HR)-associated cell death is a hallmark of plant immune responses triggered by pathogen recognition. Many effectors are known to suppress HR induced by the following two common elicitors: the pro-apoptotic mouse protein BCL2-associated X protein (BAX) and *Phytophthora infestans* elicitin inverted formin 1 (INF1) [15]. Notably, the wheat blue dwarf (WBD) phytoplasma effectors SWP12 and SWP21 (TENGU-like) were found to inhibit cell death mediated by SWP11, BAX, and/or INF1 [16]. Effectors also employ diverse mechanisms to suppress plant immunity. For instance, the apple phytoplasma effector M19_00185 acts as a ubiquitin ligase to inhibit immune activation [17], while SWP12 promotes TaWRKY74 degradation via the ubiquitin–proteasome pathway, reducing reactive oxygen species (ROS) production [18]. The JWB phytoplasma genome encodes 28 secreted proteins, which are candidate effectors with unknown functions [4]. Recent studies revealed that six non-classically secreted proteins from *Ca*. P. ziziphi effectively suppress hypersensitive cell death and H_2_O_2_ accumulation triggered by various elicitors in *N. benthamiana* [19]. However, apart from these findings, no other phytoplasma effectors linked to jujube witches’ broom disease have been reported to inhibit immune responses.

Here, we characterized *Ca*. P. ziziphi Sec-dependent effector JWB790, which suppresses BAX-induced cell death and flg22-triggered immune responses when transiently expressed in *N*. *benthamiana*. Transgenic *Arabidopsis thaliana* plants that stably express *JWB790* exhibited enhanced pathogen susceptibility. An RNA-seq analysis of these transgenic plants revealed the JWB790-mediated transcriptional reprogramming of biotic stress-responsive genes, suggesting its role in phytoplasma pathogenicity through immune signaling manipulation. Accordingly, we conclude that JWB790 is a virulence factor of *Ca*. P. ziziphi and suppresses plant immunity. This study provides new insight into effectors deployed by phytoplasma to manipulate immune responses.

## 2. Materials and Methods

### 2.1. Plant Materials and Strains

Healthy and phytoplasma-infected *Ziziphus jujuba* ‘Dongzao’ samples were collected in jujube orchards from Heze, Shandong province. Phytoplasma presence was confirmed via PCR using 16S rDNA-F/R primers [20], with sequenced products showing 100% identity to JWB phytoplasma 16S rDNA. *Arabidopsis thaliana* and *Nicotiana benthamiana* were grown in controlled chambers (23 °C, 16 h light/8 h dark cycle).

*Escherichia coli* strain DH5α was used for plasmid construction, cultured on Lysogeny Broth (LB) medium at 37 °C. *Agrobacterium tumefaciens* strain GV3101 was used for agro- infiltration and cultured in LB medium at 28 °C. *Pseudomonas syringae* pv. *tomato* (*Pst*) DC3000 was cultured on Luria-Marine (LM) medium at 28 °C.

### 2.2. Identification and Sequence Analysis of JWB790

The full-length coding sequence of the phytoplasma effector JWB790 (GenBank accession no. AYJ01076.1) was obtained from total DNA of JWB-infected plants by PCR amplification using gene-specific primers (Appendix A). The signal peptide of JWB790 was predicted using SignalP 5.0 (https://services.healthtech.dtu.dk/services/SignalP-5.0/; accessed on 14 March 2024). All the primers used in this study are described in Appendix A. Phylogenetic analysis was performed using MEGA 6.0 software with the neighbor-joining method. The phylogenetic tree was viewed by iTOL v7.

### 2.3. Alkaline Phosphatase Assay

An *E. coli* phoA gene fusion assay system was established, as previously described [21]. The *phoA* gene (GenBank NC_000913) lacking its native signal peptide (SP) sequence (designated *mphoA*) was PCR-amplified from *E. coli* strain BL21 using primers mphoA-F (with added *Bam*H I/*Hin*d III sites at the 5′-end) and phoA-R, then cloned into *Nco* I/*Xho* I-digested pET-28a (+) to generate pET-mphoA using a One-Step Cloning Kit (Vazyme, China). A full-length *phoA* construct (pET-phoA) served as a positive control. The SP sequence of *JWB790* was amplified with primers 790SP-F/R and fused in-frame to *mphoA* via *Bam*H I/*Hin*d III sites, yielding pET-790SP-mphoA. All recombinant plasmids (pET-790SP-mphoA, pET-phoA, and pET-mphoA) were individually transformed into *E. coli* BL21. Transformed cells were then plated on LB agar supplemented with 90 ng·mL^−1^ 5-bromo-4-chloro-3-indolyl phosphate (BCIP), 0.1 mM isopropyl-β-D-thiogalactopyranoside (IPTG), 75 mM Na_2_HPO_4_, and 50 μg/mL kanamycin, followed by overnight incubation at 37 °C. Colonies exhibiting blue coloration indicated successful secretion of phoA-active fusion proteins into the extracellular environment; whereas, white colonies suggested a lack of alkaline phosphatase activity. The experiment was repeated three times with similar results.

### 2.4. Agro-Infiltration Assays

For the BAX-induced cell death assay, target genes were cloned into pGR106 and transformed into *A. tumefaciens* strain GV3101, which was cultured in LB medium supplemented with 50 mg/L rifampicin and 50 mg/L kanamycin. The method for infecting tobacco leaves followed the procedure previously described [14]. Briefly, recombinant strains were cultured, pelleted, resuspended, and then infiltrated into leaves of 4-week-old *N. benthamiana* at OD_600_ = 0.2–0.4. Strains expressing GFP (negative control), Pst27791 (positive control), or JWB790 were infiltrated into leaves 24 h prior to infiltrating the strain containing BAX. Trypan blue staining was performed at 5 days post-inoculation (dpi) to visualize necrotic areas (blue-stained regions). For electrolyte leakage measurement, necrotic areas were sampled at 3 dpi using a 9 mm punch, and five leaf discs were placed in 5 mL of deionized water for 3 h at room temperature. Electrolyte leakage (E1) was measured using a conductivity meter (FE32, Mettler-Toledo, China). The samples were then boiled for 10 min, cooled, and electrolyte leakage (E2) was measured again. Conductivity percentage = (E1/E2) × 100% [22]. The experiment was repeated at least three times.

For H_2_O_2_ detection, *N. benthamiana* leaves at 3 dpi were stained with 1 mg/mL 3,3′-diaminobenzidine (DAB; DAB Color Development Kit, Coolaber, Beijing, China) and incubated under light for 8 h. The leaves were then decolorized in a 1:1 (*v*/*v*) ethanol: acetic acid solution until transparent and imaged. The experiment was repeated three times with consistent results.

For the quantitative reverse transcription PCR (qRT-PCR) analysis of defense marker genes, 100 nM flg22 was injected in *N. benthamiana* leaves expressing JWB790-GFP or GFP control at 48 h post-inoculation (hpi). Leaf samples were collected at 0 and 6 h after flg22 injection for RNA extraction. The expression of PAMP-triggered immunity (PTI)-related genes was analyzed using *NbActin* as the reference gene [23]. Three independent biological replicates, each with three technical replicates, were performed for all samples.

### 2.5. Generation and Analysis of Transgenic A. thaliana Lines and Bacterial Inoculation Assay

Transgenic *A. thaliana* lines were generated using the floral dip method [24]. Briefly, the *A. tumefaciens* strain GV3101 harboring the 35S:*JWB790*-GUS construct was cultured, pelleted, and resuspended in infection medium (1/2 MS media supplemented with 10 mM MES (pH 5.7), 5% (*w*/*v*) sucrose, 0.2 mM acetosyringone, and 0.05% (*v*/*v*) Silwet L-77(Coolaber)). After 2–3 h of preconditioning, *A. thaliana* inflorescences were dipped for 30 s, maintained under dark/high-humidity conditions for 24 h, and then transferred to greenhouse conditions (23 °C, 16-h light/8-h dark cycle) for seed maturation. Transgenic lines were confirmed by both PCR and β-glucuronidase (GUS) histochemical staining. For GUS staining, the GUS enzyme catalyzes the hydrolysis of X-Gluc (5-bromo-4-chloro-3-indolyl-β-D-glucuronide), producing an insoluble blue precipitate. Three-week-old seedlings were incubated with 20 μg/mL X-Gluc at 37 °C for 8 h (GUS Stain Kit, Coolaber), followed by decolorization and imaging.

Bacteria inoculation assay was performed as previously described [25]. Rosette leaves from 4- to 5-week-old homozygous T_3_ plants and wild-type (WT) were then infiltrated with *Pst* DC3000 at a concentration of OD_600_ = 0.001–0.002. At 3 dpi, representative leaves were rinsed, dried, and cut into discs, with four discs from two leaves forming one biological repeat. Discs were ground, mixed with sterilized water, and serially diluted. Dilutions were plated on rifampicin-supplemented LM solid medium at 28 °C for 12–18 h. Three technical replicates were performed per sample, and the plates were air-dried before bacterial quantification. Bacterial colonization was calculated as colony-forming units (CFU)/cm^2^ and represented on a logarithmic scale. The experiment was repeated three times with independent biological replicates. Defense-related gene expression was analyzed by qRT-PCR using *AtActin* as the internal control gene [26]. Three biological and technical replicates were performed for each sample.

Callose deposition assay and ROS staining were performed according to the methods previously described [13], with a slight difference. Briefly, three 3-week-old *A. thaliana* leaves from each line were vacuum infiltrated with 10 μM flg22. After 24 h, the leaves were treated with absolute ethyl alcohol: acetic acid (1:1 *v*/*v*) to remove the chlorophyll. The cleared leaves were stained with 0.1% (*w*/*v*) aniline blue in 150 mM K_2_HPO_4_ (pH 8.5/KOH) in the dark for 1 h. Callose deposits were captured under ultraviolet light using a fluorescence microscope. The number of callose deposits in each image were counted using ImageJ2 software ([27]. To assess ROS accumulation, 3-week-old seedlings were treated with 10 μM flg22 for 4 h, then stained with 1 mg/mL DAB (DAB Color Development Kit, Coolaber). The samples were then decolorized and imaged. The experiment was repeated three times with similar results.

### 2.6. RNA Isolation and Quantitative Reverse Transcription-PCR (qRT-PCR) Analysis

Total RNA was extracted using the Quick RNA Isolation Kit (Huayueyang Biotechnology, Beijing, China) following the manufacturer’s protocol. First-strand cDNA was synthesized from 2 µg of RNA using the RevertAid First-Strand cDNA Synthesis Kit (Thermo Fisher Scientific, Waltham, MA, USA). qRT-PCR was performed on a CFX Connect Real-Time System (Bio-Rad, Hercules, CA, USA) with gene-specific primer pairs listed in Appendix A. Gene expression levels were quantified using the comparative 2^-∆∆Ct^ method. Three technical repeats were included for each qRT-PCR analysis. Statistical significance was determined using an unpaired two-tailed Student’s *t*-test.

### 2.7. RNA-Seq Analysis

For RNA-seq analysis, we collected leaf, stem, and flower tissues from 8-week-old T_3_ generation JWB790-overexpressing transgenic *Arabidopsis* plants for equal-quantity pooled sampling, with three independent transgenic lines serving as biological replicates. As controls, wild-type *Arabidopsis* plants of the same age were sampled identically, also with three biological replicates. The transcriptome sequencing process involved several critical steps to ensure high-quality data generation and analysis. First, RNA integrity was assessed using the Bioanalyzer 2100 system to confirm sample quality. For library preparation, mRNA was purified from total RNA using poly-T oligo-attached magnetic beads. The following two types of libraries were constructed: non-strand-specific and strand-specific. The non-strand-specific library involved cDNA synthesis with random hexamer primers, followed by end repair, adapter ligation, size selection, and amplification. The strand-specific library incorporated dUTP during second-strand synthesis and included USER enzyme digestion for directional sequencing. Libraries were quantified using Qubit and real-time PCR, and size distribution was verified with the Bioanalyzer. After quality control, libraries were pooled and sequenced on an Illumina platform using the “Sequencing by Synthesis” principle, where fluorescent signals were captured and converted into sequence data. Bioinformatics analysis began with raw data processing using Fastp to remove adapters, poly-N reads, and low-quality sequences. Clean reads were mapped to a reference genome using Hisat2, which accounts for splice junctions. Novel transcripts were predicted with StringTie (Version 2.2.1), and gene expression levels were quantified using featureCounts and FPKM. Differential expression analysis was performed with DESeq2 or edgeR, followed by GO and KEGG enrichment analyses to identify functional pathways.

## 3. Results

### 3.1. JWB790 Is a Sec-Dependent Effector and Expressed in JWB-Infected Tissues

To elucidate the virulence and molecular mechanisms of the jujube witches’ broom phytoplasma, we analyzed the genome of the isolate jwb-nky. JWB790 (GenBank accession number: AYJ01076.1) is a lysine-rich effector, which encodes a 121-amino acid (aa) protein containing a putative N-terminal signal peptide (790SP; Figure 1a). To verify Sec-dependent secretion of JWB790, an *Escherichia coli* alkaline phosphatase (phoA) gene fusion assay was conducted (Appendix A). On indicator medium, *E. coli* cells containing pET-mphoA remained white after 12 h of incubation, while those carrying pET-790SP-mphoA turned blue as the positive control (Figure 1b), indicating that 790SP successfully directed mphoA to translocate outside of the bacteria. These findings confirm JWB790 as a canonical Sec-secreted effector.

Phylogenetic analysis revealed that JWB790 clusters with apple phytoplasma effector M19_00185, a known ubiquitin ligase involved in immunity suppression (Figure 1c). qRT-PCR analysis showed that *JWB790* was upregulated in JWB-infected tissues, including chlorotic leaves and phyllody structures (Figure 1d), suggesting its role in *Ca*. P. ziziphi pathogenesis. The presence of *Ca*. P. ziziphi in these samples was verified by 16S ribosomal DNA cloning (Appendix A). The presence of JWB790 in the infected tissues was detected by PCR (Appendix A).

### 3.2. JWB790 Suppresses Cell Death and Basal Immune Response in Nicotiana benthamiana

One criteria for assessing the ability to inhibit plant immunity is the suppression of programmed cell death (PCD) [28,29]. Since *Ca*. P. ziziphi is not amenable for transformation, we evaluated the ability of JWB790 to suppress BAX (Bcl-2-associated X protein)-induced PCD, which resembles the defense-related hypersensitive response (HR) in plant cells. Similar to Pst27791 [14], which served as the positive control here, JWB790 effectively suppressed BAX-induced PCD in *N. benthamiana* (Figure 2a). Necrotic areas were visualized using trypan blue staining (Figure 2b). We further quantified the cell death by electrolyte leakage assays. The results showed that the electrolyte leakage of *N. benthamiana* leaves injected with JWB790/BAX was significantly decreased compared with that co-injected with GFP/BAX (Figure 2d). Because ROS is a crucial trigger of cell death, we further tested if JWB790 suppresses BAX-triggered H_2_O_2_ accumulation. The result showed that DAB staining in the leaf regions infiltrated with JWB790/BAX was notably weaker than those infiltrated with GFP/BAX (Figure 2c). These results demonstrate that JWB790 suppresses BAX-triggered cell death and H_2_O_2_ accumulation.

To further test the ability of JWB790 to suppress plant immunity, we analyzed the immune response triggered by bacterial flagellar peptide flg22 in *N. benthamiana*. *NbPR1a* and *NbWRKY33* are marker genes that are responsive to PAMPs in *N. benthamiana* [23]. The results showed that the overexpression of *JWB790* suppressed flg22-triggered expression of *NbPR1a* and *NbWRKY33* (Figure 2e,f). Together, these results suggest that JWB790 inhibits both elicitor-induced cell death and basal immunity in *N. benthamiana*.

### 3.3. Overexpression of JWB790 in Arabidopsis thaliana Reduces Disease Resistance

To determine whether JWB790 promotes pathogen infection in plants, we generated three T_3_ transgenic *A. thaliana* plants that stably express *JWB790* under the control of the 35S promoter and tested their resistance to *Pst* DC3000. *JWB790* expression in transgenic lines was confirmed by PCR (Appendix A), and protein expression was verified by GUS staining (Appendix A). Compared with wild-type (WT) plants, the disease resistance of *JWB790* transgenic *A*. *thaliana* lines was significantly reduced (Figure 3a). *Pst* DC3000 colonization in *JWB790* transgenic *A*. *thaliana* lines was significantly higher than that in control plants at 3 dpi (Figure 3b and Appendix A). Additionally, qRT-PCR analysis showed that the transcript levels of marker genes for salicylic acid (SA) and jasmonic acid (JA) signaling pathways, including *PR1* (pathogenesis-related protein 1), *LOX3* (lipoxygenase 3), and *ICS1* (isochorismate synthase 1), were significantly reduced in *JWB790* transgenic plants at 48 h post-inoculation (hpi) (Figure 3c–e).

In addition, the transgenic plants showed a significant decrease in H_2_O_2_ production triggered by flg22, as demonstrated by the DAB staining results (Figure 3f,g). Furthermore, aniline blue staining revealed a significant reduction in callose deposition in the flg22-infiltrated leaves of transgenic plants compared to those of WT Col-0 (Figure 3h,i). Taken together, these findings demonstrate that JWB790 reduces plant defense responses and facilitates pathogen infection in *A*. *thaliana.*

### 3.4. JWB790 Reprograms the Expression of Biotic Stimulus-Related Genes

To elucidate the impact of JWB790 on biological processes in transgenic *Arabidopsis* plants, we conducted a comprehensive RNA-seq analysis, which identified 2120 differentially expressed genes (DEGs), including 672 downregulated and 1448 upregulated genes (Appendix A and Appendix A). The subsequent GO and KEGG enrichment analyses of these DEGs between WT and JWB790 transgenic *Arabidopsis* lines revealed significant functional insights. Multiple GO terms related to biological processes and molecular functions were enriched, particularly ‘secondary metabolic process’, ‘hydrolase activity’, and ‘protein serine/threonine kinase activity’ (Figure 4a). Moreover, KEGG pathway analysis identified the following three major categories of significantly enriched metabolic pathways: (1) energy metabolism, including pyruvate metabolism and glycolysis/gluconeogenesis; (2) secondary metabolism, particularly phenylpropanoid biosynthesis and glucosinolate biosynthesis; and (3) stress response mechanisms, such as plant–pathogen interaction and the MAPK signaling pathway in plants (Figure 4b).

Functional annotation of the DEGs indicated that the expression profiles of various genes related to abiotic and biotic stresses were affected in the *JWB790*-overexpressing transgenic *Arabidopsis* lines. The expression of key regulatory factors in immune signaling pathways, including the ethylene response factors (ERFs) *ERF38* and *ERF39*, was downregulated. Similarly, the expression of genes associated with secondary metabolism, such as CYP706A5 (a cytochrome P450 enzyme critical for phytoalexin synthesis) and CAD9 (a cinnamyl alcohol dehydrogenase required for lignin biosynthesis), was reduced (Figure 4c). Notably, three redox homeostasis-related DEGs, including two peroxidases (PODs, AT5G64120 and AT3G61770) and one ascorbate peroxidase (APX), were also suppressed in *JWB790*-overexpressing transgenic lines (Figure 4c). Intriguingly, several genes known to negatively regulate plant immunity were significantly upregulated (Figure 4d). These included the transcription factor ERF109, which suppresses JA signaling and disrupts ROS accumulation [30], and WRKY38, a negative regulator of pathogenesis-related (PR) gene expression [31]. Overall, the RNA-seq analysis demonstrated that the overexpression of *JWB790* led to a reprogramming of biotic stimulus-related gene expression, potentially contributing to the enhanced susceptibility of the transgenic plants.

## 4. Discussion

Phytoplasma pathogens exhibit a unique life cycle that distinguishes them from other well-characterized plant pathogenic bacteria. Their evolutionary adaptation to dual-kingdom parasitism—spanning plant hosts and insect vectors—represents a remarkable ecological strategy [32]. This dual-host capability implies the evolution of sophisticated molecular mechanisms that enable the manipulation of essential cellular processes in phylogenetically distinct eukaryotic hosts. In our study, the transcript expression of *JWB790* was detected in JWB-infected tissues (Figure 1d and Appendix A), suggesting its crucial role in *Ca*. P. ziziphi pathogenicity. However, the expression dynamics of *JWB790* in insect vectors remain unclear. Notably, no phytoplasma effector has yet been demonstrated to suppress immunity in both hosts, making this an underexplored yet ecologically significant research frontier.

Phytoplasmas defy the conventional pathogenesis paradigm of Gram-negative bacterial pathogens. Typical plant-pathogenic bacteria maintain extracellular colonization through apoplastic spaces and deploy type III secretion system (T3SS)-delivered effectors [10,33]. In contrast, phytoplasmas exhibit an obligate intracellular lifestyle, inhabiting the cytoplasm of living sieve cells. This niche specialization likely drives their genomic reduction (e.g., loss of T3SS/T4SS machinery) and reliance on the Sec-dependent secretion pathway, as demonstrated by the functional annotation of SecA homologs in phytoplasma genomes [5,32]. Our study demonstrates that JWB790 encodes a secreted protein containing a functional N-terminal signal peptide, as confirmed by phoA fusion assays (Figure 1b). This finding not only confirms its Sec system dependency but also raises intriguing questions about intercellular trafficking mechanisms, such as how do these cytoplasmic effectors translocate across sieve cell membranes and traverse plant tissues? It is hypothesized that these proteins are transported through the plasmodesmata that connect plant cells, but the specific mechanisms remain unclear [12].

Plants defend against pathogens through a multilayered process, including PAMP-triggered immunity (PTI) and effector-triggered immunity (ETI). Both share downstream signaling pathways, including the production of ROS, callose deposition, and increased levels of defense-related hormones [34]. Research has demonstrated that phytoplasma infection triggers ROS accumulation and callose deposition in host plants [35,36]. Our study demonstrated that the overexpression of *JWB790* in *Arabidopsis* plants significantly suppresses flg22-triggered ROS production and callose deposition (Figure 3f–i). Transcriptomic profiling of *JWB790*-overexpressing *Arabidopsis* plants revealed the substantial downregulation of key redox regulators, including PODs and APX (Figure 4c). Furthermore, the transcription factor *ERF109*, which suppresses JA signaling and disrupts ROS accumulation, was upregulated (Figure 4d). These findings provide compelling evidence that JWB790 plays a crucial role in modulating plant immune responses. Interestingly, the mechanism by which plants detect phytoplasmas to initiate immune responses remains unclear. Unlike typical bacteria, phytoplasmas lack both an outer cell wall and flagella. It is assumed that intracellular proteins CSPs and EF-Tu may induce PTI [37]. The precise mechanism underlying phytoplasma-triggered plant immunity warrants further investigation.

In summary, JWB790 exemplifies how phytoplasma effectors subvert plant immunity. However, several knowledge gaps remain. The direct host targets of JWB790 have yet to be identified. Addressing these questions will not only deepen our understanding of JWB790’s role in pathogenicity but also inform the development of durable resistance strategies against phytoplasma-associated diseases. For instance, future applications of spray-induced gene silencing (SIGS) using siRNA to selectively silence *JWB790* in jujube leaves may establish a novel foundation for disease control.

## 5. Conclusions

This study demonstrates that JWB790, a Sec-dependent effector from *Ca*. P. ziziphi, plays a critical role in suppressing plant immunity and promoting pathogenicity. Through functional assays in *N. benthamiana* and *A. thaliana*, we confirmed that JWB790 inhibits programmed cell death, reduces ROS accumulation, suppresses callose deposition, and downregulates the expression of defense marker genes, thereby compromising plant defense responses. *JWB790*-overexpressing transgenic *Arabidopsis* plants exhibited enhanced susceptibility to pathogens. Transcriptome analysis revealed that JWB790 reprograms the expression of genes involved in defense, secondary metabolism, and stress responses, further compromising plant immunity. These findings provide valuable insights into the molecular mechanisms of JWB disease and highlight JWB790 as a potential target for developing strategies to control phytoplasma-associated plant diseases.

## Figures and Tables

**Figure 1 biology-14-00528-f001:**
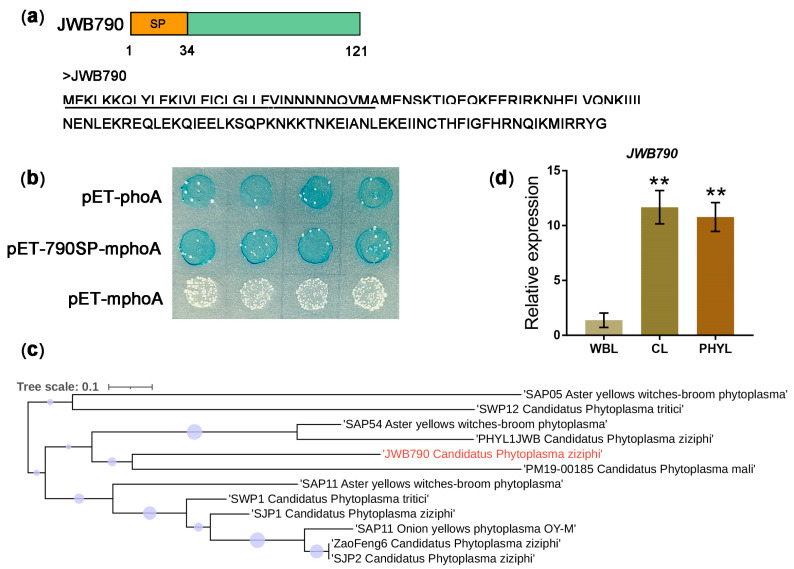
Secretion and expression analysis of JWB790. (**a**) Schematic diagram and amino acid sequence of the JWB790. Signal peptide of JWB790 (790SP) is indicated by a black line. (**b**) The 790SP directed the extracellular translocation of the mphoA lacking its native SP. On indicator Lysogeny Broth (LB) medium containing 90 μg/mL BCIP, 1 mM IPTG (induce *lacUV5* promoter and *T_7_lac* promoter), 50 μg /mL Kanamycin, and 75 mM Na_2_HPO_4_ (suppress the endogenous phosphatase activity), the *Escherichia coli* cells expressing the fusion protein 790SP-phoA turned blue after 12 h of incubation at 37 °C. (**c**) Phylogenetic analysis of JWB790. A phylogenetic tree was constructed with MEGA 6.0 software using the neighbor-joining method. Bootstrap values (percentage) are represented by circles at branch points, where larger circles indicate higher values. JWB790 is marked as the red texts. (**d**) Transcript level of *JWB790* in jujube witches’ broom (JWB) infected tissues. Relative expression was calculated using the comparative threshold (2^−ΔΔCT^) method. The qRT-PCR values were normalized to the transcript level of *JwbTuf* (CP025121.1:478755-479918). Transcript levels in witches’ broom leaves were standardized to 1. Mean ± SE (standard error), *n* = 3; **, *p* < 0.01 compared with the witches’ broom leaves; unpaired two-tailed Student’s *t*-test. CL: chlorotic leaves. WBL: witches’ broom leaves. PHYL: phyllody.

**Figure 2 biology-14-00528-f002:**
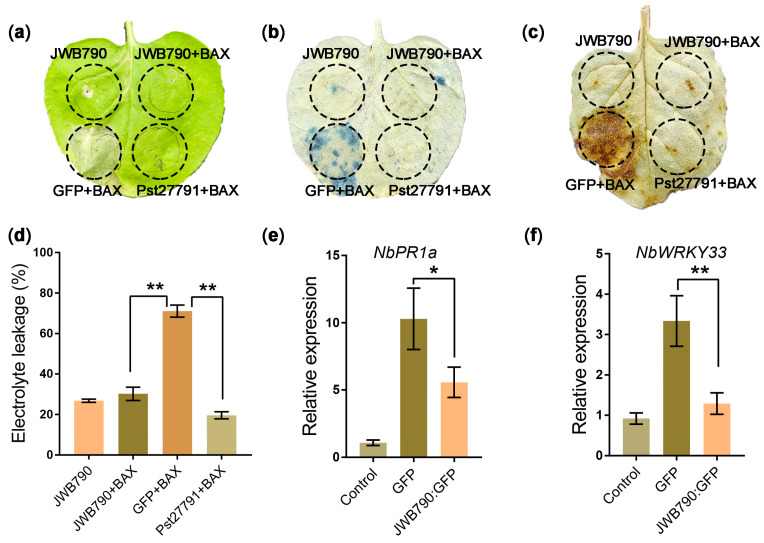
JWB790 suppresses BAX-triggered cell death and flg22-induced plant immune responses. (**a**) JWB790 suppresses BAX-triggered cell death. *Nicotiana benthamiana* leaves were infiltrated with *Agrobacterium tumefaciens* carrying potato virus X (PVX) *PVX-Pst27791* (a *Puccinia striiformis* f. sp. *tritici* effector served as the positive control here), *PVX-GFP*, or *PVX-JWB790*, respectively. After 24 h, *A. tumefaciens* carrying *BAX* was infiltrated into the right half of the leaf. The leaf was photographed 5 days post-infiltration (dpi) of BAX. (**b**) Visualization of necrotic areas at 5 dpi by trypan blue staining (blue indicates cell death). Images were captured after destaining. (**c**) H_2_O_2_ production in *N*. *benthamiana* leaves was detected by DAB staining. Leaves were infiltrated with *A. tumefaciens* harboring the indicated constructs and stained with 1 mg/mL DAB at 3 dpi. Images were taken after destaining. (**d**) Quantification of cell death by electrolyte leakage. The statistical analyses were performed with Student’s *t*-test. Bars indicate mean ± SE. ** *p* < 0.01. The experiments were repeated three times with similar results. (**e**,**f**) JWB790 suppresses the expression of defense marker genes. The transcript levels of *NbPR1a* and *NbWRKY33* in the leaves transiently expressing *GFP-JWB790* and *GFP* (control) were determined by qRT-PCR after infiltration with 100 nM flg22. Leaf samples were collected at 0 and 6 hpi with flg22. The value at 0 hpi (control) was standardized to 1. *NbActin* was used as the reference gene (mean ± SE, *n* = 3; *, *p* < 0.05; **, *p* < 0.01; unpaired two-tailed Student’s *t*-test). The experiments were repeated three times with similar results.

**Figure 3 biology-14-00528-f003:**
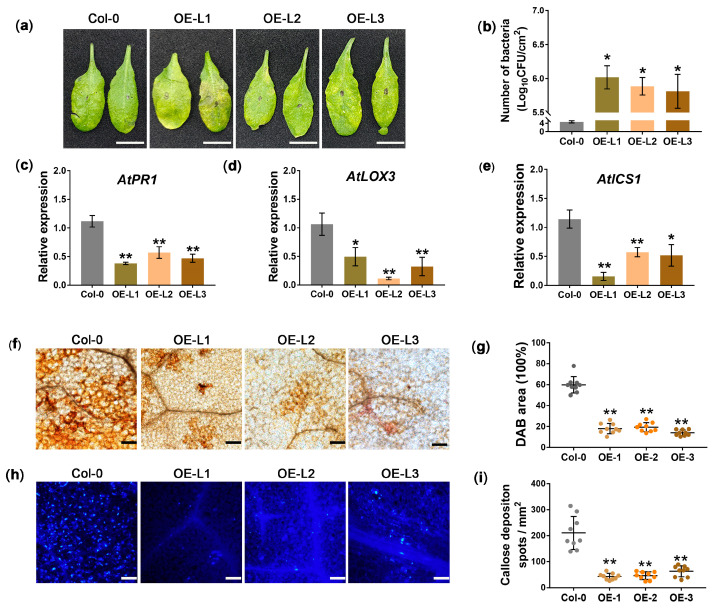
JWB790 enhances the susceptibility of transgenic *Arabidopsis thaliana.* (**a**) Phenotype of *Pst* DC3000-infected transgenic *A. thaliana* lines at 3 days post-infiltration (dpi). (**b**) JWB790 promotes the colonization of *Pst* DC3000 in transgenic *A. thaliana* plants. Leaves of transgenic lines and wild-type Col-0 were inoculated with *Pst* DC3000 suspensions. Bacterial colonization was determined as colony-forming units (CFU)/cm^2^ at 3 dpi. Bars represent mean ± SE from three biological replicates (six leaves per sample). * *p* < 0.05. The experiments were repeated three times with similar results. (**c**–**e**) Relative expression of *AtPR1, AtLOX3* and *AtICS1* in *JWB790* transgenic *A. thaliana* at 48 hpi with *Pst* DC3000 was detected by qRT-PCR. *AtActin* was used as the internal reference (mean ± SE. * *p* < 0.05; ** *p* < 0.01; unpaired two-tailed Student’s *t*-test). Experiments were repeated three times with similar results. (**f**) H_2_O_2_ production was decreased in *JWB790* transgenic *A. thaliana*. Leaves were stained with DAB 4 h after infiltration with 10 μM flg22 and imaged under bright-field microscopy. Scale bar = 100 μm. (**g**) Quantification of DAB-stained area (%) per mm^2^ using ImageJ2. Data represent mean ± SE from nine sites (three biological replicates). ** *p* < 0.01 (unpaired two-tailed Student’s *t*-test). (**h**) Callose deposition was reduced in *JWB790* transgenic *A. thaliana*. Leaves were stained with aniline blue 24 h after 10 μM flg22 infiltration and imaged under UV microscopy. Scale bar = 100 μm. (**i**) Statistical analysis of the callose deposits per mm^2^ were conducted using ImageJ2. Means were based on the calculation of nine sites from three biological replicates (mean ± SE. ** *p* < 0.01; unpaired two-tailed Student’s *t*-test).

**Figure 4 biology-14-00528-f004:**
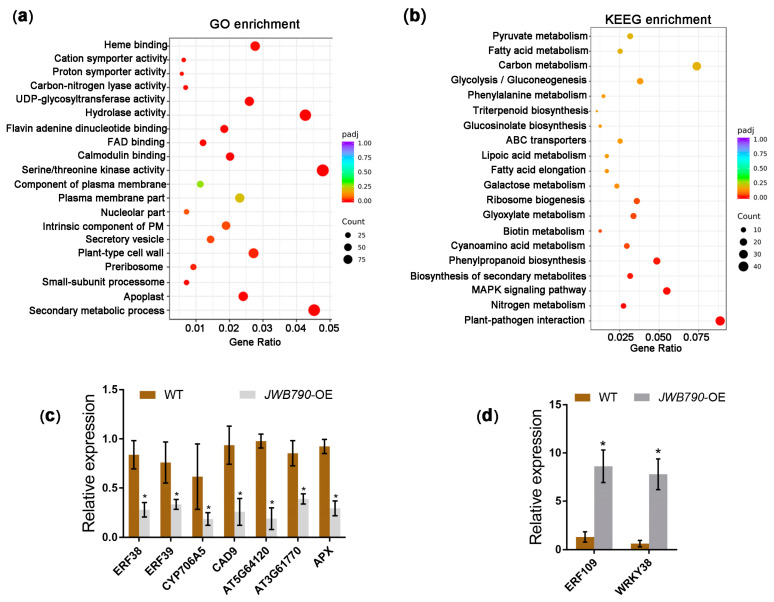
JWB790 reprograms the expression of biotic stimulus-related genes in transgenic *Arabidopsis* plants. (**a**) GO enrichment analysis of the differentially expressed genes (DEGs) between JWB790 transgenic *Arabidopsis* lines and wild-type (WT). The x-axis shows the ratio of DEGs annotated to a GO term relative to the total number of DEGs. The y-axis lists the GO term. Dot size corresponds to the number of genes annotated to each GO term, and the color gradient indicates the enrichment significance level. (**b**) KEGG enrichment analysis of the DEGs between *JWB790* transgenic *Arabidopsis* lines and WT. The x-axis shows the ratio of DEGs annotated to a KEGG pathway relative to the total number of DEGs. The y-axis lists the KEGG pathways. Dot size represents the number of genes annotated to each pathway, and the color gradient reflects the enrichment significance. (**c**) Downregulation of selected DEGs (mean ± SE. * *p* < 0.05; unpaired two-tailed Student’s *t*-test). (**d**) Upregulation of selected DEGs (mean ± SE. * *p* < 0.05; unpaired two-tailed Student’s *t*-test).

## Data Availability

Data are contained within the article or Appendix A.

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
