# Peer review of "A Sec-Dependent Effector from “Candidatus Phytoplasma ziziphi” Suppresses Plant Immunity and Contributes to Pathogenicity"

_biology, 2025, doi:10.3390/biology14050528_

Round 1

Reviewer 1 Report

Comments and Suggestions for Authors

The manuscript of Wan et al., entitled “A SEC-dependent effector from “Candidatus Phytoplasma ziziphi” suppresses plant immunity and contributes to pathogenicity” represents a consistent contribution in the understanding of how phytoplasmas succeed to colonize the host by overcoming plant defense mechanisms.

The research was well conducted, the paper is written clearly with just a few issues. Figures are well executed as well as those in the supplementary data.

Comments and suggestions:

Abstract

Line 16

Cell death should be “programmed cell death”

Lines  16-17

“When JWB790 was introduced into Arabidopsis thaliana plants or Pseudomonas syringae pv. tomato (Pst) DC3000, it increased disease severity and reduced resistance to pathogens.”

What should the readers understand by reading this? How can a protein introduced in Arabidopsis, or in a bacterium, have similar results like “increased disease severity and reduced resistance to pathogens.” Maybe in Arabidopsis but not in Pseudomonas! A much better writing of what actually has been achieved can be found in lines 93-95.

For clarity, the authors should rewrite this sentence.

Line 28

“implicated” should be “was found to be implicated”

Line 32

Use simple past tense like “revealed” rather than present “reveal”. Same in lines 35 (increased), 36 (reduced).

Line 40

Decrease should be “decreases”

Line 52

Ca. P. ziziphin should be Ca. P. ziziphin, that is, removed the n.

Lines 109 and 112

The medium to grow Pst DC3000 is quite unusual. Pst is usually grown on King's B medium or Nutrient Yeast Glycerol (NYG) medium. Why using Luria-Marine for Pst? Any particular  reason?

Line 196

Citation 22 is not correct. See https://imagej.net/software/imagej/

The correct citation should be either the link or

Schneider, C. A., Rasband, W. S., & Eliceiri, K. W. (2012). NIH Image to ImageJ: 25 years of image analysis. Nature Methods9(7), 671–675. doi:10.1038/nmeth.2089

The paper referred as citation 22 also wrongly cites another paper which also has an incomplete citation (“with ImageJ software (NIH)”), which is sad!

  1. Xu, Q.; Tang, C.; Wang, X.; Sun, S.; Zhao, J.; Kang, Z.; Wang, X. An Effector Protein of the Wheat Stripe Rust Fungus Targets Chloroplasts and Suppresses Chloroplast Function. Nat. Commun. 2019, 10, doi:10.1038/s41467-019-13487-6.

Lines 224 to 242

This section is written like in a report/general info from a sequencing/processing platform or a teaching resource, using present tense. Everything should be re-written using simple past tense.

For example “The transcriptome sequencing process involves several critical steps to ensure high-quality data generation and analysis. First, RNA integrity is assessed using the Bioanalyzer 2100 system to confirm sample quality.” should be written something like “The transcriptome sequencing process involved several critical steps to ensure high-quality data generation and analysis. First, RNA integrity was assessed using the Bioanalyzer 2100 system to confirm sample quality.”

Comments on the Quality of English Language

English is OK

Author Response

Comments 1: Line 16, Cell death should be “programmed cell death”

Response: Revised.

Comments 2: Lines 16-17

“When JWB790 was introduced into Arabidopsis thaliana plants or Pseudomonas syringae pv. tomato (Pst) DC3000, it increased disease severity and reduced resistance to pathogens.”

What should the readers understand by reading this? How can a protein introduced in Arabidopsis, or in a bacterium, have similar results like “increased disease severity and reduced resistance to pathogens.” Maybe in Arabidopsis but not in Pseudomonas! A much better writing of what actually has been achieved can be found in lines 93-95.

For clarity, the authors should rewrite this sentence.

Response: Thank you for your suggestion. We revised this sentence as “Transgenic Arabidopsis thaliana plants that stably express JWB790 exhibit enhanced pathogen susceptibility” (Line 17-18).

Comments 3: Line 28, “implicated” should be “was found to be implicated”

Response: Revised.

Comments 4: Line 32

Use simple past tense like “revealed” rather than present “reveal”. Same in lines 35 (increased), 36 (reduced).

Response: Revised.

Comments 5: Line 40, Decrease should be “decreases”

Response: Revised.

Comments 6: Line 52, Ca. P. ziziphin should be Ca. P. ziziphin, that is, removed the n.

Response: Revised.

Comments 7: Lines 109 and 112

The medium to grow Pst DC3000 is quite unusual. Pst is usually grown on King's B medium or Nutrient Yeast Glycerol (NYG) medium. Why using Luria-Marine for Pst? Any particular reason?

Response: Thanks for your question. This study selected Luria-Marine (LM) medium for culturing Pst DC3000 based on the following considerations:

First, we followed the standardized protocol published by Yuan and Xin (2021) in Bio-Protocol (Citation 26: Bacterial Infection and Hypersensitive Response Assays in Arabidopsis-Pseudomonas syringae Pathosystem. 2021, 11, 1-11, doi:10.21769/BioProtoc.4268.), which explicitly used LM medium for the Arabidopsis-Pseudomonas syringae interaction system. Second, compared to commonly used King's B medium (rich in glycerol and phosphates) or NYG medium (high in organic nitrogen sources), The nutrient-limiting conditions of LM medium may more effectively induce the expression of bacterial pathogenicity-related genes. Given that this research focuses on plant defense response mechanisms, employing the LM medium may allow better simulation of natural infection conditions, thereby yielding biologically meaningful results.

Recipes for Luria-Marine (LM) medium in this study: Tryptone 10 g/L, Yeast extract powder 6 g/L, KH2PO4 1.5 g/L, NaCl 0.6 g/L, MgSO4 0.35 g/L. Dissolve the ingredients in distilled water, adjust the pH to 7 with sodium hydroxide (NaOH) solution, and autoclave the solution.

Comments 8: Line 196

Citation 22 is not correct. See https://imagej.net/software/imagej/

The correct citation should be either the link or

Schneider, C. A., Rasband, W. S., & Eliceiri, K. W. (2012). NIH Image to ImageJ: 25 years of image analysis. Nature Methods9(7), 671–675. doi:10.1038/nmeth.2089

The paper referred as citation 22 also wrongly cites another paper which also has an incomplete citation (“with ImageJ software (NIH)”), which is sad!

  1. Xu, Q.; Tang, C.; Wang, X.; Sun, S.; Zhao, J.; Kang, Z.; Wang, X. An Effector Protein of the Wheat Stripe Rust Fungus Targets Chloroplasts and Suppresses Chloroplast Function. Nat. Commun. 201910, doi:10.1038/s41467-019-13487-6.

Response: Revised. See citation 22: Line 541-542.

Comments 9: Lines 224 to 242

This section is written like in a report/general info from a sequencing/processing platform or a teaching resource, using present tense. Everything should be re-written using simple past tense.

For example, “The transcriptome sequencing process involves several critical steps to ensure high-quality data generation and analysis. First, RNA integrity is assessed using the Bioanalyzer 2100 system to confirm sample quality.” should be written something like “The transcriptome sequencing process involved several critical steps to ensure high-quality data generation and analysis. First, RNA integrity was assessed using the Bioanalyzer 2100 system to confirm sample quality.”

Response: Thanks for your keen observation in identifying this oversight. We have carefully revised the paragraph to address this issue and ensure the accuracy of our manuscript (Line 210-228).

Reviewer 2 Report

Comments and Suggestions for Authors

This manuscript identified an effector JWB790 from a phytoplasma causing Jujube Witches’ Broom. The authors showed JWB790 is an immune-suppressing virulence effector, by transient expression in Nicotiana benthamiana. Also, they demonstrated JWB790 is a virulence effector by stable expression in the model plant Arabidopsis as well as transformation in the model phytopathogenic bacterium PstDC3000. Further, they investigated the impact of JWB790 on plant biological processes by RNA-seq analysis. Overall, the experiments were well-designed, and results gained novel insights into phytoplasma-plant interaction.  

My major concern:

When testing the virulence role of JWB790 in N. benthamiana, the authors inoculated PstDC300 after JWB790 transient expression. I am doubt this experiment design, because N. benthamiana is generally considered as a non-host of PstDC300 (see “A Pseudomonas syringae pv. tomato DC3000 mutant lacking the type III effector HopQ1-1 is able to cause disease in the model plant Nicotiana benthamiana”, DOI: 10.1111/j.1365-313X.2007.03126.x). The necrotic area observed in Figure 3b and chlorosis in Figure 3d may resulte from hypersensitive response triggered by PstDC300, not disease caused by this bacterium. Because the authors have already gained sufficient evidence for JWB790 virulence, I suggest deleting Figure 3, or else they can inoculate other bacteria like Pseudomonas syringae pv. tabaci and Ralstonia solanacearum.

Minor concerns:

  1. Line 12, “phytoplasmas” should be singular.
  2. Line 74, it should be “especially those of obligate parasites”.
  3. Line 92, “flg22-triggered …”
  4. Line 117, delete “by” or “using”.
  5. Line159,the authors described 100 nM flg22 was used, but
  6. Line 344, Schematic diagram of…

Author Response

 Major concern:

Comments 1:When testing the virulence role of JWB790 in N. benthamiana, the authors inoculated PstDC300 after JWB790 transient expression. I am doubt this experiment design, because N. benthamiana is generally considered as a non-host of PstDC300 (see “A Pseudomonas syringae pv. tomato DC3000 mutant lacking the type III effector HopQ1-1 is able to cause disease in the model plant Nicotiana benthamiana”, DOI: 10.1111/j.1365-313X.2007.03126.x). The necrotic area observed in Figure 3b and chlorosis in Figure 3d may resulte from hypersensitive response triggered by PstDC300, not disease caused by this bacterium. Because the authors have already gained sufficient evidence for JWB790 virulence, I suggest deleting Figure 3, or else they can inoculate other bacteria like Pseudomonas syringae pv. tabaci and Ralstonia solanacearum.

Response: Thank you for your kind suggestion. We have removed this part of the results and made corresponding deletions throughout the entire manuscript (including the Simple Summary, Abstract, Introduction, Materials and Methods, Discussion, Conclusion and Supplementary Materials) to ensure consistency.

Minor concerns:

Comments 1: Line 12, “phytoplasmas” should be singular.

Response: Revised.

Comments 2: Line 74, it should be “especially those of obligate parasites”.

Response: Revised.

Comments 3: Line 92, “flg22-triggered …”

Response: Revised.

Comments 4: Line 117, delete “by” or “using”.

Response: Revised.

Comments 5: Line159,the authors described 100 nM flg22 was used, but

Response: Revised.

Comments 6: Line 344, Schematic diagram of

Response: Revised. Thanks for your keen observation in identifying this oversight. We rewrite the sentences. Line 250: (a) Schematic diagram and amino acid sequence of the JWB790. Line 464: Figure S1: Schematic diagram of the prokaryotic expression cassettes for the phoA, mphoA and 790sp-mphoA.